# Expression of Cellular and Extracellular *TERRA*, *TERC* and *TERT* in Hepatocellular Carcinoma

**DOI:** 10.3390/ijms23116183

**Published:** 2022-05-31

**Authors:** Michele Manganelli, Ilaria Grossi, Jessica Corsi, Vito Giuseppe D’Agostino, Katarina Jurikova, Emilio Cusanelli, Sarah Molfino, Nazario Portolani, Alessandro Salvi, Giuseppina De Petro

**Affiliations:** 1Department of Molecular and Translational Medicine, Division of Biology and Genetics, University of Brescia, 25123 Brescia, Italy; m.manganelli@unibs.it (M.M.); ilaria.grossi@unibs.it (I.G.); 2Department of Cellular, Computational and Integrative Biology, CIBIO, University of Trento, 38123 Trento, Italy; jessica.corsi@unitn.it (J.C.); vito.dagostino@unitn.it (V.G.D.); katarina.jurikova@unitn.it (K.J.); emilio.cusanelli@unitn.it (E.C.); 3Department of Clinical and Experimental Sciences, Surgical Clinic, University of Brescia, 25123 Brescia, Italy; sarahmolfino@gmail.com (S.M.); nazario.portolani@unibs.it (N.P.)

**Keywords:** hepatocellular carcinoma, telomeric-repeat-containing RNAs (*TERRA*), *TERC*, *TERT* mRNA, extracellular vesicles

## Abstract

Non-coding RNAs are transcribed from telomeres and the telomeric repeat-containing RNAs (*TERRA*) are implicated in telomere homeostasis and in cancer. In this study, we aimed to assess in hepatocellular carcinoma (HCC) the cellular and extracellular expression of *TERRA*, the telomerase RNA subunit (*TERC*) and the telomerase catalytic subunit (*TERT*). We determined by qPCR the expression level of *TERRA* 1_2_10_13q, *TERRA* 15q, *TERRA* XpYp, *TERC* and of *TERT* mRNA in HCC tissues and in the plasma of HCC patients. Further, we profiled the same transcripts in the HCC cell lines, HA22T/VGH and SKHep1C3, and in the extracellular vesicles (EVs) derived from their secretomes. We found that the expression of *TERRA* and *TERT* mRNA was significantly deregulated in HCC, being *TERRA* downregulated and *TERT* mRNA upregulated in HCC tissues vs. the peritumoral (PT) ones, and the receiver operating characteristic (ROC) curve analyses revealed a significant ability in discriminating HCC from PT tissue. Further, the determinations of circulating *TERRA* and *TERC* showed higher amounts of these transcripts in the plasma of HCC patients vs. controls and ROC analyses gave significant results. The expression characterization of the cultured HCC cells showed their ability to produce and secrete *TERRA* and *TERC* into the EVs; the ability to produce *TERT* mRNA that was not detectable in the EVs; and the ability to respond to sorafenib treatment increasing *TERRA* expression. Our results highlight that: (i) both cellular and extracellular expressions of *TERRA* and *TERC* are dysregulated in HCC as well as the cellular expression of *TERT* mRNA and (ii) the combined detection of *TERRA* and *TERC* in plasma may represent a promising approach for non-invasive diagnostic molecular indicators of HCC.

## 1. Introduction

The transcriptional activity of the human genome (>90% of the genome) produces long non-coding RNAs (lncRNAs) whose functions are mostly unknown. The extremities of linear chromosomes, or telomeres, actively synthesize a heterogeneous population of lncRNAs named *TERRA* (telomeric repeat-containing RNA) with various sizes (from 100 bases to 10 Kb) and containing the canonical telomeric repeat sequence UUAGGG, as well as sequences unique to the subtelomeric region of each chromosome [1,2]. *TERRA* transcripts localize at telomeres and interact with the telomerase RNA subunit (*TERC*) and with the telomerase catalytic subunit (*TERT*) [3,4]. Several studies have established the role of *TERRA* in telomere homeostasis and its implication in cancer. In this regard, *TERRA* has been shown to sustain the activation of DNA damage response pathways at dysfunctional telomeres [5,6] and to facilitate completion of telomere DNA replication in cancer cells [7]. Importantly, *TERRA* depletion associates with an increase of telomerase activity, supporting the notion that *TERRA* can negatively regulate telomerase in mammalian cells [2,3,4]. Expression studies conducted by various technologies have revealed the dysregulation of *TERRA* in various solid tumors, a frequent downregulation in specific tumor types vs. their normal tissue counterparts [8], but also its overexpression in medulloblastoma [9]. Moreover *TERRA* expression is increased upon the treatment of human cancer cell lines with chemotherapeutic agents, suggesting that *TERRA* transcripts may participate in the cellular response and/or resistance to chemotherapy [10,11,12]. For hepatocellular carcinoma (HCC), it is known that telomerase activation is a hallmark of this malignant condition [13] and that *TERRA* may participate in telomerase regulation in this cancer type [8]. HCC is the most common type of primary liver cancer and worldwide is the fifth cancer and the third cause of cancer mortality; it is a highly heterogeneous disease and it occurs most often in people with chronic liver diseases (liver hepatitis, alcohol hepatitis, viral B or C hepatitis, steatohepatitis, cirrhosis). Also for this reason, the HCC clinical management remains difficult: very often the diagnosis reveals an advanced HCC that usually is not submitted to surgical resection and a resectable HCC frequently leads to recurrent tumors [14,15,16,17]. To date, the treatment options are limited and advancing in the molecular characterization of HCC is a real challenge. Also, early diagnosis represents a critical step for the management of HCC and the identification of novel biomarkers of HCC will be of foremost importance.

The aim of the current study was to characterize *TERRA* expression in HCC, determining by qPCR its expression levels in HCC tissues and in HCC cell lines as well as its circulating amount in the plasma of HCC patients and in the cell secretomes. Further, we assessed in the same biological specimens the expression of *TERC* and of *TERT* mRNA in relation to *TERRA*. Our results highlight that both cellular and extracellular expressions of *TERRA* and *TERC* are dysregulated in HCC, as well as the cellular expression of *TERT* mRNA.

## 2. Results

### 2.1. Dysregulated Expression of TERRA, TERC and TERT mRNA in HCC Tissues

To characterize the expression of *TERRA* in HCC patients, we first examined *TERRA* levels in paired HCC and peritumoral tissues by RT qPCR. *TERRA* expression from different telomeres, analyzed using specific primer pairs (as described in the Materials and Methods, *TERRA*1 = 1_2_10_13q; *TERRA*2 = 15q; *TERRA*3 = XpYp), resulted to be significantly downregulated (Appendix A). *TERRA* expression was obtained as a mean of the different *TERRA* relative quantifications and, as shown in Figure 1a by the qPCR data, *TERRA* was significantly downregulated in HCC tissues compared with their matched peritumoral tissues (RQ_HCC_ average vs. RQ_PT_ average, *p*-value = 0.02).

The R values refer to the ratio RQ_HCC_/RQ_PT_ and, among the HCC tested, 19/25 of cases (76%) showed downregulation of *TERRA* expression, whereas 6/25 cases (24%) showed its upregulation (Figure 1b). No correlation was observed between *TERRA* expression and the clinicopathological characteristics of HCC patients (Appendix A). ROC curve analyses, generated for the different *TERRA* subpopulations considered, using the primer pairs mentioned above, revealed that each *TERRA* target had a significant capability in discriminating HCC from peritumoral tissues. Further logistic regression model of classifiers showed that the use of the combined *TERRA* molecules might represent a significant successful discriminating tool to distinguish HCC from peritumoral tissue (Figure 1c and Table 1) (AUC: 0.79; CI: 0.563–0.829; *p*-value = 0.0382).

Given the importance of telomerase in HCC, we also investigated the expression of *TERC* and *TERT* mRNA in the paired HCC and peritumoral tissues by qPCR. For *TERC* expression, the data obtained displayed a trend of upregulation in HCC tissues compared with the PT ones (Figure 2a, RQ_HCC_ average vs. RQ_PT_ average, *p*-value = ns) and receiver operating characteristic (ROC) curve analysis did not disclose a significant capability in distinguishing HCC from PT tissues (Figure 2c) (AUC:0.51; CI: 0.34–0.67; *p*-value = 0.89). 

*TERT* mRNA expression (Figure 3a) was significantly upregulated (*p*-value = 0.019) at different orders of magnitude in HCC tissues compared to the PT ones (Figure 3b). ROC curve analysis of *TERT* mRNA expression revealed a successful significant ability in discriminating HCC from PT tissues (Figure 3c) (AUC: 0.716; CI: 0.57–0.86; *p*-value = 0.008). These findings suggest that quantification of *TERRA* and *TERT* mRNA may represent a discriminating tool to distinguish HCC from peritumoral tissue.

### 2.2. Determination of Circulating TERRA and TERC in the Plasma of HCC Patients 

Since *TERRA* has been found in human blood plasma and in the culture medium of human cell lines [18,19], to characterize further *TERRA* expression in HCC, we measured its amount in the plasma of HCC patients and in healthy individuals; due to the lack of a known reference gene that would allow the normalization of circulating *TERRA*, we report qPCR data as the average raw cycle thresholds (Ct) in plasma samples [20,21]. As shown in Figure 4 and Table 2, circulating *TERRA*1 and *TERRA*3 amounts, corresponding to *TERRA* transcripts expressed from telomeres 1_2_10_13q (*TERRA*1) and telomeres XpYp (*TERRA*3), were significantly higher in the plasma of HCC patients compared to healthy subjects (Figure 4a,c, *p*-values = 0.0023 and 0.031, respectively) and *TERRA*2 amounts (corresponding to the 15q transcripts) revealed a trend of upregulation (Figure 4b, *p*-value = ns). 

No correlation was observed between circulating levels of *TERRA* and the clinicopathological characteristics of HCC patients (Appendix A). ROC curve analyses were generated for different *TERRA* subpopulations to inquire as to their capability to act as tools to successfully distinguish HCC from healthy subjects. In particular, the logistic regression model of *TERRA* classifiers evidenced the significant combined use of *TERRA* targets (Figure 4d and Table 2) (AUC: 0.765; CI: 0.624–0.873; *p*-value = 0.0004). To support these findings, we calculated the average Ct values of *TERRA* in the 2 groups of individuals and we obtained significantly higher levels in plasma from HCC patients in comparison to healthy control subjects (Appendix A, *p*-value = 0.038). Further, we evaluated the correlation by Spearman’s analysis of *TERRA* levels between the HCC tissues and the plasma samples from the same patients. The R-value was 0.3096 and the *p*-value was not significant (*p* value = 0.1410). Thus, in our cohort there was no correlation between *TERRA* levels in HCC solid biopsies and liquid biopsies (Appendix A). Further, as shown in Figure 5, the ddPCR determination of *TERC* in plasma revealed a significantly higher level in HCC patients vs. control subjects (2.118 vs. 0.826 copies/μL, *p*-value = 0.0019); and the ROC analysis gave an AUC value = 0.8120 with a *p*-value = 0.0002. On the contrary, *TERT* mRNA was not detectable in liquid biopsy specimens (data not shown). These findings suggest that the combined detection of circulating *TERRA* and *TERC* may represent a promising approach for the diagnosis of HCC.

### 2.3. Characterization of Cellular and Extracellular TERRA in HCC Cell Lines 

Considering the evidence provided by the present study on the altered expression of *TERRA*, *TERC* and *TERT* mRNA in HCC tissues and on the detection of *TERRA* and *TERC* in the plasma of HCC patients, we aimed to explore the cellular and extracellular expression of these transcripts in human HCC cell lines in order to gain mechanistic insights on the function of *TERRA* in HCC. Since *TERRA* transcripts may participate in the cellular response to anticancer compounds [10,11,12], first we assessed the transcript expression in HA22T/VGH and SKHep1C3 cells (Figure 6) and monitored the possible expression variation following an anticancer drug treatment, in particular the sorafenib treatment in sensitive and resistant cells (sf and SR cells, respectively) [22,23]; this is because sorafenib is one of the drugs used by clinicians for the treatment of advanced HCC [14].

As shown in Figure 6a, the SK-Hep1 C3 cells express a very high amount of *TERRA* compared to HA22T/VGH cells (RQ_SKHep1C3_ = 67.63 vs. RQ_HA22T/VGH_ = 1.03; fold increase, FI = 65.9) and a lower amount of *TERC* and *TERT* mRNA compared to HA22T/VGH cells (Figure 6b,c). These cell lines are both undifferentiated HCC cells and can be considered as models of the heterogeneity of HCC; in particular, the HA22T/VGH cells might represent a model of the HCC set with *TERRA* down-modulation and the SKHep1C3 may be a model of HCC with *TERRA* upregulation. Both cell lines responded similarly to sf treatment, enhancing *TERRA* expression and with the FI higher in the HA22T/VGH cells which are characterized by a low basal level (FI*_TERRA_* = 7.70 and FI*_TERRA_* = 1.91 in HA22T/VGH and SK-Hep1C3 cells, respectively). The expression variation does not seem to depend on the sensitivity or resistance of the cells to sf, since this effect of the drug appears similar in both conditions. Both cell lines also responded similarly to sf treatment enhancing *TERC* and *TERT* mRNA expression, for *TERC* with a low FI (FI*_TERC_* = 1.3 and FI*_TERC_* = 1.14 in HA22T/VGH and SKHep1C3 cells, respectively) and for *TERT* mRNA with a higher FI in HA22T/VGH cells (FI*_TERT_* = 5.01 in HA22T/VGH cells and a FI*_TERT_* = 1.3 in the SKHep1C3 cells). We then isolated extracellular vesicles (EV) from the secretome of both cell lines to determine the amounts of extracellular transcripts detectable in the EV fraction. For *TERRA*, the results shown in Figure 6d revealed that: (i) *TERRA* was detectable in the EVs of both HCC cell lines, the amount of *TERRA* ascertained in the EVs of HA22T/VGH cells being very high, and (ii) both cell lines displayed a lower amount of *TERRA* in the EVs of sf-treated cells, thus showing an opposite trend of expression variation compared to the intracellular one. The *TERRA* size detected in the EVs was in a size range of less than 500 nt as revealed by Northern analysis (Appendix A). For *TERC*, it was detectable (Figure 6e) in the EVs derived from HA22T/VGH cells and, in this case, sf treatment led to expression variations similar to the cellular one, that is, an increase of the *TERC* amount. *TERC* was not detectable in the EVs derived from SK-Hep1C3 cells even when ddPCR was employed to determine its amount and the EVs of both HCC cell lines did not display *TERT* mRNA molecules. In particular, these findings reveal that the HCC cells expressing low *TERRA* levels (HA22T/VGH) show a very high *TERRA* amount into the EVs. Nevertheless, in both cell lines the sf treatment leads to a cellular *TERRA* increase and at the same time causes a *TERRA* decrease in the EVs.

## 3. Discussion

The expression variations of the lncRNA *TERRA* in cancer has generated a growing interest on the biology of these telomeric transcripts in various solid tumors (larynx and colon cancer, medulloblastoma, glioblastoma, head and neck cancer, cervical and endometrial carcinoma) [2,9,24,25,26,27,28], but its real functional role in a given cancer type remains to be elucidated. In the present work, for the first time, we determined *TERRA* expression in HCC by qPCR, in HCC tissues and in the plasma of HCC patients. We used a qPCR assay that did not detect all *TERRA* transcripts, but it detected the most abundant ones. In particular, we used three primer pairs to quantify *TERRA* expression levels transcribed from seven telomeres (*TERRA*1: 1_2_10_13q; *TERRA*2: 15q; *TERRA*3: XpYp) [29]. Furthermore, Diman et al. [30] demonstrated that the primer pair named *TERRA*1 (1_2_10_13q) detected the transcripts expressed from the telomeres: 1_2_4_10_13_22q. Thus, according to Diman et al., our qPCR assay should recognize *TERRA* transcripts expressed from nine telomeres. Furthermore, Montero et al. [31] demonstrated that *TERRA* is mainly transcribed from the 20q subtelomere in human cells and the primer pairs able to detect this transcript were not included in our qPCR assay. Moreover, considering the role of *TERRA* in the telomere function and the importance of the telomerase enzyme in the biology of HCC, in the present study, we also assessed the RNA expression of the main components of the telomerase complex, the RNA subunit *TERC* and the telomerase catalytic subunit *TERT* mRNA. We found a significant *TERRA* downregulation in HCC tissues vs. the corresponding PT ones and a significantly higher amount of *TERRA* in the plasma of HCC patients. In both cases, for the first time, ROC curves indicated significant capabilities in distinguishing HCC tissues from PT ones and HCC patients from control subjects. Further, the qPCR determinations of *TERRA* downregulation in HCC tissues, consistent with the in situ hybridization data reported by Cao et al. [8], could be utilized as specific characterization of the malignant tissue (of HCC or other cancer types), being the qPCR technique frequently used in research laboratories.

For *TERRA* transcripts in the plasma, an RNA-seq study [18] revealed the presence of *TERRA* molecules in the plasma of 2 normal subjects and of 19 patients with various cancer types (prostate, stomach, breast, colon, kidney, lung, duct, liver, melanoma, ovarian); however, to our knowledge, there are no reports on circulating *TERRA* in a small or large cohort of patients with the same type of cancer, or even reports on the PCR determinations of *TERRA* amounts in the plasma of cancer patients. Hence, the data reported in our study are the first concerning *TERRA* expression in the plasma of a cohort of patients with the same cancer type, HCC, in comparison to healthy individuals.

For the overexpressed amount of *TERRA* detected in the plasma of HCC patients, we cannot know the origin of the circulating forms of *TERRA*, if they derive from the malignant cells or not. Despite this, our data indicate that the plasma determination of *TERRA* amounts can be taken into account as a possible non-invasive molecular indicator of HCC diagnosis and, in order to ascertain this hypothesis, we are planning a multicenter study to examine a larger cohort of patients.

For *TERC* and *TERT* mRNA expression we found an upregulation in HCC tissues, which was significant for the *TERT* mRNA. For measuring the level of *TERC* and *TERT* mRNA in plasma, we used ddPCR, the best technology to assess absolute quantifications of transcripts, especially the circulating ones. The *TERC* amounts detected in the plasma of HCC patients were significantly higher, while *TERT* mRNA was not detectable, thus indicating that in HCC patients the cells do not secrete the detectable extracellular form of *TERT* mRNA. The ROC analysis of circulating *TERC* expression revealed a significant ability in discriminating HCC patients from controls. Concerning *TERC* in cancer, it is known from in situ and fluorescence hybridization data (ISH, FISH) that is overexpressed in lung, oral, prostate and cervix carcinoma [10]; further, only Novakovic et al., to our knowledge, reported nested-PCR data that revealed the presence of *TERC* in the plasma of healthy controls and of patients with breast cancer, melanoma and thyroid cancer, but did not allow the *TERC* amount to be considered as a tumor marker [29]. Hence, our findings would indicate the determination of circulating *TERRA* and *TERC* in HCC patients as specific non-invasive molecular indicators of this malignant condition.

Subsequently, in the present work, to address the problem of the dysregulation of *TERRA*, *TERC* and *TERT* mRNA in HCC from a biological point of view, we characterized two HCC cell lines for the expression of these transcripts and, for the first time, we report data on their expression both at the cellular and extracellular levels, indicating that the two HCC cell lines may represent the heterogeneity of HCC. In particular, the HA22T/VGH and SKHep1C3 cells, at low and very high *TERRA* expressions, respectively, may represent the HCC cases with *TERRA* downregulation and upregulation, respectively. HA22T/VGH cells producing low *TERRA* amounts expressed high levels of *TERC* and *TERT* mRNA, and this result is consistent with previous evidence indicating that *TERRA* depletion associates with increased levels of *TERC* [3]. Both HCC cell lines secrete *TERRA* molecules into the secretome-derived EVs. The HA22T/VGH cells secreted very high amounts of *TERRA* and low amounts of *TERC*. As expected, *TERT* mRNA was not detected as an extracellular form in any condition. Further, concerning the effect of an anticancer drug on *TERRA* expression, both cell lines similarly responded to sorafenib treatment enhancing the expression amount of *TERRA*, *TERC* and *TERT* mRNA at the cellular level. HA22T/VGH reacted to sf increasing the cellular expression of *TERRA*, *TERC* and *TERT* mRNA and decreasing the secretion of *TERRA* and *TERC*; SKHep1C3 cells reacted to sf increasing the cellular expression of *TERRA*, but much less *TERC* and *TERT* mRNA, and decreasing the secretion of *TERRA* and by not secreting both *TERC* and *TERT* transcripts. Sorafenib is a multikinase inhibitor exerting its antitumor activity mainly affecting the proliferation and apoptosis of cancer cells, known to influence gene expression at the level of mRNAs, ncRNAs and genome methylation with mechanisms that have not been elucidated [22,23,32]. Hence, further effort is needed to understand the intracellular events that led to an increase of *TERRA* in cells treated with sf. From a general point of view, it is known that *TERRA* is a heterogeneous group of ncRNAs generated from multiple different telomeres and with various size (from 100 to 10 Kb); it has been proposed that its upregulation can be regulated by stress inducing the processing of *TERRA* molecules into smaller fragments then secreted into the EVs. It seems that *TERRA* containing EVs are relatively stable and circulate in extracellular spaces, including plasma, but it is not known how the smaller extracellular forms are generated. As reported by Wang et al. [19], the extracellular molecules of *TERRA* can be breakdown products of full-length cellular *TERRA* or an alternative form that is encapsulated into the EVs and secreted in extracellular spaces. It is likely that *TERRA* containing EVs can transfer signaling information to surrounding cells impacting the extracellular microenvironment. The complex combination of factors that comprise EVs and the type of recipient cells sensing the EVs may determine the nature of the signaling and response.

Hence, this part of the present study is the first effort to characterize the cellular and extracellular expression of *TERRA*, *TERC* and *TERT* mRNA in HCC cell lines demonstrating the capability of these cells to secrete *TERRA* and *TERC* transcripts into the EVs and thus opening a new scenario in the biology of HCC cells. Novel issues will be addressed to understand the role of *TERRA* and *TERC* containing EVs in transferring signaling to surrounding cells in vitro and in an HCC tumor xenograft.

In conclusion, in the current work we have assessed the cellular and extracellular expression of *TERRA*, *TERC* and *TERT* mRNA studying HCC tissues, the plasma of HCC patients and the cells and secretomes of two HCC cell lines. Our novel findings obtained by qPCR show that HCC tissue is significantly characterized by *TERRA* downregulation and by *TERT* mRNA upregulation; the plasma of HCC patients revealed an overexpression of *TERRA* and *TERC*, thus indicating novel non-invasive indicators of HCC whose clinical diagnostic significance will be investigated further in a larger cohort of patients. It is clear that among HCC tissues there are two sets, one with *TERRA* down-modulation, the other with *TERRA* upregulation. Moreover, the characterization of two HCC cell lines, representative of this feature of HCC heterogeneity, has clearly shown the ability of HCC cells to produce and secrete *TERRA* and *TERC* into the EVs; the ability to produce *TERT* mRNA that is not secreted; and the ability to respond to sorafenib treatment increasing *TERRA* expression. From a general point of view, our findings contribute to the basic knowledge of the cellular and extracellular forms of *TERRA* and *TERC* produced by cultured cancer cell lines and those detectable in HCC tissues and in the plasma of HCC patients.

## 4. Materials and Methods

### 4.1. Tissue Samples and Liquid Biopsies Collection

All human HCC tissues as well as the corresponding peritumoral (PT) (resected 1–2 cm from the malignant tumor) and the peripheral blood were obtained from 25 HCC patients who underwent HCC resection at Spedali Civili, Surgical Clinic of Brescia, Italy (Appendix A). The peripheral blood of healthy volunteers (*n* = 25) was collected at the Immunohematology and Transfusion Medicine Service at ASST Civili Hospital in Brescia, Italy. HCC tissues and the PT ones were subsequently preserved in RNA-Later (Invitrogen; Thermo Fisher Scientific, Inc., Waltham, MA, USA) for further investigation. The subjects consisted of 20 men and 5 women, ranging from 60 to 82 years of age. The subjects did not have any apparent distant metastases, and none of them had been previously treated for HCC. Each individual was screened for the presence of the hepatitis B virus (HBV) or hepatitis C virus (HCV): 3 patients were positive for HBV, 15 were positive for HCV and 8 were found to be negative for both HBV and HCV. Each biopsy specimen underwent a pathological examination as previously described [22] that revealed the presence of different background diseases: 3 steatosis, 14 cirrhosis, 6 chronic hepatitis, 1 von Meyenburg complex and 1 no apparent background pathology (normal hepatic parenchyma); for 1 sample, no information was available. Liquid biopsies were collected from the peripheral blood of each patient before HCC surgical resection in a VACUETTE^®^ (Greiner Bio-One, S.r.l., Kremsmünster, Austria) EDTA-coated blood collection tubes. Plasma samples were obtained from 1 mL of peripheral blood with a first centrifugation step at 3000 rpm for 10 min at 4 °C followed by a second centrifugation step at 4000 rpm for 20 min at 4 °C. Samples showing hemolysis were excluded. The plasma was transferred to a new tube and stored at −80 °C until RNA extraction.

This study was approved by the ethical committee of ASST Civili Hospital of Brescia on 2 October 2012 (NP1230). Informed consent was obtained from all the subjects enrolled in this study. All methods were performed in accordance with the relevant guidelines and regulations.

### 4.2. Cell Cultures, Treatment with Sorafenib and EV Isolation

In the present study, human tumor cell lines derived from HCC (HA22T/VGH and SKHep1C3) were used. HA22T⁄VGH undifferentiated cells, kindly provided by N. D’Alessandro (University of Palermo, Palermo, Italy), were maintained in RPMI-1640 (Life Technologies; Thermo Fisher Scientific, Inc., Waltham, MA, USA) with 100 nM sodium pyruvate (Thermo Fisher Scientific, Inc., Waltham, MA, USA). SKHep1Clone3 (SKHep1C3), selected from human HCC-derived cells (SKHep1: ATCC HTB-52), was maintained in Earle’s MEM (Thermo Fisher Scientific, Inc., Waltham, MA, USA). All culture media were supplemented with 10% fetal bovine serum (FBS; Euroclone, S.p.a., Pero, Italy) and 10,000 U/mL penicillin/streptomycin (Thermo Fisher Scientific, Inc., Waltham, MA, USA). HA22T/VGH and SKHep1C3 sorafenib resistant cells (HA22T/VGH-SR and SKHep1C3-SR), produced in our lab, were maintained as previously described [22]. Cells were cultured at 37 °C with 5% CO_2_ in 10 cm Ø plates up to almost 90% of confluence. Sorafenib [N-(3-trifluoromethyl-4-chlorophenyl)-N-(4-(2-methylcarbamoylpyridin-4-yl) oxyphenyl) urea] was synthesized and provided by Bayer Corporation (West Haven, CT, USA). The compound was dissolved in 100% dimethyl sulfoxide (DMSO; Sigma-Aldrich; Merck, Inc., Darmstadt, Germany) and diluted with RPMI-1640 or MEM to the required concentration (10 µM, 15 µM). In in vitro experiments, 0.1% DMSO was added to cultures as a negative solvent control. In order to get extracellular vesicles (EVs)-containing media (named in this text as secretome), HA22T/VGH and SKHep1C3 cells were seeded in a 10 cm Ø plate (4 replicates/each condition) at a density of 5 × 10^6^ cells and maintained in complete media until they reached 75% of confluence. Then, after gentle washes with PBS, cells were subsequently supplied with serum-free medium for 24 h and treated with sorafenib at the specific concentrations (15 µM).

Before starting the nickel-based isolation (NBI) of EV, collected media were centrifuged at 2800× *g* for 10 min to remove cell debris and mixed to a nickel-based matrix of beads in order to capture EVs, which were subsequently analyzed by Q-NANO instrument (IZON Science, Ltd., Christchurch, New Zealand), as previously reported [33]. We recovered a particle concentration range between 1.63 × 10^8^ and 2.09 × 10^8^ EV/mL for HA22T/VGH cells, from 3.08 × 10^8^ to 11.01 × 10^8^ EV/mL for SKHep1C3 cells according to different cell densities and the amount of medium processed to further optimize the RNA recovery. The EV showed various sizes of the mean diameter, from 185 to 252 nm for HA22T/VGH cells and from 250 to 317 nm for the SKHep1C3 ones.

### 4.3. RNA Isolation

Total RNA was obtained from 10 cm Ø cells plates and by grinding ~1 cm^3^ of tumor and peritumor tissue with a TissueRupture (Qiagen, Inc., Hilden, Germany) in 1 mL of TRIzol Lysis Reagent (Qiagen, Inc., Hilden, Germany) according to the Chomczynski–Sacchi method [34]. RNA purity and concentration were assessed and measured by a NanoDrop™ 1000 Spectrophotometer (Thermo Fisher Scientific, Inc., Waltham, MA, USA), while the integrity was evaluated on 0.8% agarose gel stained with ethidium bromide. Extracellular total RNA was extracted from 200 µL of isolated EVs and 200 µL of plasma, using a miRNeasy Mini Kit (Qiagen, Inc., Hilden, Germany), according to the manufacturer’s instructions as previously reported [20].

### 4.4. Northern Blotting

The secretome RNA samples were extracted as described above, the control total RNA from HeLa (ATCC: CCL-2) and U2OS (ATCC: HTB-96) cell lines was extracted using TRIzol reagent (Thermo Fisher Scientific, Inc., Waltham, MA, USA) according to the manufacturers’ protocol.

U2OS (human osteosarcoma cell line) and HeLa (human cervical adenocarcinoma cell line) cells were obtained from ATCC (Manassas, VA, USA). The cells were grown in Dulbecco’s Modified Eagle Medium (Thermo Fisher Scientific, Inc., Waltham, MA, USA) supplemented with 2 mM L-glutamine, 1% penicillin-streptomycin and 10% fetal bovine serum at 37 °C and 5% CO_2_ to 50–70% confluency before harvesting for RNA extraction.

RNA purity and concentrations were assessed as stated in Section 4.3. RNA was mixed with RNA gel loading dye (Thermo Fisher Scientific, Inc., Waltham, MA, USA), denatured for 10 min at 65 °C and separated for 2 h at 9.5 V/cm in 1% agarose MOPS gel with 1.9% formaldehyde, then blotted onto the Amersham Hybond-N+ hybridization membrane (Cytiva) with the semi-dry Power Blotter System (Invitrogen; Thermo Fisher Scientific, Inc., Waltham, MA, USA) and cross-linked in the CL-1000 Ultraviolet Crosslinker (Fisher Scientific, Inc., Hampton, NH, USA) at the pre-set conditions. A probe consisting of five telomeric repeats (5′-CCCTAA-3′)5 was labeled with the DIG Oligonucleotide 3′-End Labeling Kit (F. Hoffmann-La Roche, Ltd.; Basel, Switzerland) according to the manufacturers’ instructions and denatured for 5 min at 95 °C before hybridization. The hybridization was performed at 50 °C overnight, the membrane was washed, blocked, incubated with anti-digoxygenin-AP (F. Hoffmann-La Roche, Ltd.; Basel, Switzerland; dilution 1:1500), visualized by incubation with CDP-Star Merck KGaA, Darmstadt, Germany and captured on the Chemidoc XRS+ (Bio-Rad Laboratories, Inc., Hercules, CA, USA).

### 4.5. RT-qPCR and Gene Expression Analysis

cDNA for *TERC* and *TERT* mRNA studies was synthesized from 1 μg of tissues and cells total RNA using M-MLV reverse transcriptase (Thermo Fisher Scientific, Inc., Waltham, MA, USA) with random hexamer (Thermo Fisher Scientific, Inc., Waltham, MA, USA), according to the manufacturer’s instructions. The reverse transcription (RT) reaction was performed at 37 °C for 60 min, followed by inactivation at 95 °C for 5 min in a T100 Thermal Cycler (Bio-Rad Laboratories, Inc., Hercules, CA, USA). *TERRA* first-strand cDNA was synthesized following Feretzaki’s protocol [35]. Briefly, 1 μg of tissues and cells total RNA was first partially denatured at 65 °C before RT. Then, a *TERRA*-cDNA-pool was synthesized in a final volume of 20 µL with RT-specific primers: lncRNA *TERRA* 5′-(CCCTAA)5-3′ and β-actin 5′-AGTCCGCCTAGAAGCATTTG-3′. cDNA for circulating lncRNA *TERRA* was synthesized from 2.5 µL of plasma total RNA using the iScript Explore One-Step RT and PreAmp Kit (Bio-Rad Laboratories, Inc., Hercules, CA, USA) in a final reaction volume of 25 µL, according to the manufacturer’s instructions. The pre-amplification step consisted of 14 cycles of temperature variation at 95 °C for 15 s and 60 °C for 4 min.

qPCR for *TERC* and *TERT* mRNA were carried out using TaqMan Universal PCR Master Mix 2X (Applied Biosystems; Thermo Fisher Scientific, Inc., Waltham, MA, USA) and the appropriate qPCR assays 20X, following the manufacturer’s instructions: *TERC* (assay ID: ID Hs03454202_s1; Thermo Fisher Scientific, Inc., Waltham, MA, USA), *TERT* (assay ID: Hs00972650_m1; Thermo Fisher Scientific, Inc., Waltham, MA, USA). GAPDH was used as endogenous control (assay ID: Hs99999905_m1; Thermo Fisher Scientific, Inc., Waltham, MA, USA). The thermocycling conditions were: initial denaturation at 95 °C for 10 min, followed by 40 cycles at 95 °C for 15 s then 60 °C for 1 min. qPCR for lncRNA *TERRA* was performed mixing 2 µL of *TERRA*-cDNA-pool (1:2 from tissues or cells) or 3 µL of *TERRA*-cDNA-pool (1:10 of pre-amplified product) and the specific forward/reverse primers with Power SYBR Green PCR Master Mix 2X (Applied Biosystems; Thermo Fisher Scientific, Inc., Waltham, MA, USA) in a final volume of 20 µL, according to the manufacturer’s instructions. *TERRA* qPCR was followed by the dissociation stage for melting curve analysis for each primer-pair used at 95 °C for 25 min, 60 °C for 1 min and 95 °C for 15 min. The sequence of the specific primers (Integrated DNA Technologies, Inc.) was: *TERRA* 1 (hTel_1_2_10_13q) 5′-GAATCCTGCGCACCGAGAT-3′ (forward) 5′-CTGCACTTGAACCCTGCAATAC-3′ (reverse); *TERRA* 2 (hTel_15q) 5′-CAGCGAGATTCTCCCAAGCTAAG-3′ (forward) 5′-AACCCTAACCACATGAGCAACG-3′ (reverse); *TERRA* 3 (hTel_XpYp) 5′-GCAAAGAGTGAAAGAACGAAGCTT-3′ (forward) 5′-CCCTCTGAAAGTGGACCAATCA-3′ (reverse). β-actin was used as endogenous control 5′-CCTCGCCTTTGCCGATCC-3′ (forward), 5′-GGATCTTCATGAGGTAGTCAGT-3′ (reverse). Each reaction was performed in triplicate. Data analysis was performed by the 7500 Real-Time PCR System (Applied Biosystems; Thermo Fisher Scientific, Inc., Waltham, MA, USA). In tissues and cells, gene expression was measured with the Livak–Schmittgen (2-ΔΔCq) method [36] and the *TERRA* expression was obtained as the mean of the relative quantification (RQ) values for *TERRA* 1, *TERRA* 2 and *TERRA* 3. When all 25 patients were analyzed, the plates were virtually merged by the 7500 Applied Biosystems software in order to determine the same calibrator per target (*TERRA*1, *TERRA*1, *TERRA*3) that was the sample with the lowest *TERRA* expression (in our cohort the sample HCC347). For all these reasons, RQ values of the different samples can be compared.

Due to the lack of a known reference gene that would allow the normalization of circulating *TERRA*, qPCR data were reported as the average raw cycle thresholds (Ct) in plasma samples.

### 4.6. ddPCR and Gene Expression Analysis

Total RNA from plasma was reverse-transcribed into cDNA by random hexamers using M-MLV reverse transcriptase (Thermo Fisher Scientific, Inc., Waltham, MA, USA). Then, 9 μL of the resulting cDNA was prepared for amplification in a 20 μL reaction volume containing 10 μL 2X ddPCR Supermix for probes (Bio-Rad Laboratories, Inc., Hercules, CA, USA) and 1 μL 20X TaqMan (Thermo Fisher Scientific, Inc., Waltham, MA, USA) PCR probe assay (*TERT*: assay ID Hs00972650_m1; *TERC*: assay ID Hs03454202_s1). Each ddPCR assay mixture (20 μL) was loaded into a disposable droplet generator cartridge (Bio-Rad Laboratories, Inc., Hercules, CA, USA). Then, 70 μL of droplet generation oil for probes (Bio-Rad Laboratories, Inc., Hercules, CA, USA) was loaded into each of the eight oil wells. The cartridge was then placed inside the QX200 droplet generator (Bio-Rad Laboratories, Inc., Hercules, CA, USA). When droplet generation was completed, the droplets were transferred to a 96-well PCR plate (Bio-Rad Laboratories, Inc., Hercules, CA, USA) using a multichannel pipette. The plate was heat-sealed with foil and placed in a conventional thermal cycler. Thermal cycling conditions were: 95 °C for 10 min, then 40 cycles of 94 °C for 30 s and 60 °C for 1 min (ramping rate reduced to 2%), with a final step at 98 °C for 10 min and a 4 °C indefinite hold. A no template control (NTC) and a negative control for each reverse transcription reaction (RT-neg) were included in every assay.

### 4.7. Statistical Analysis

Statistical analysis was carried out using GraphPad Prism v7.0 (GraphPad Software, Inc., San Diego, CA, USA) software. The Wilcoxon test was used to compare the expression levels of selected transcripts between peritumoral and tumoral tissues. An unpaired *t*-test was used to test the significant differences in the plasma levels of selected transcripts between HCC patients and healthy subjects and for in vitro results. Diagnostic performance was evaluated using receiver operating characteristic (ROC) curve analysis. To assess the diagnostic values of multi-assays, the logistic regression method was applied. The diagnostic performance of individual as well as combinations of classifiers was computed using MedCalc v19.1.7 (MedCalc Software, Ltd., Ostend, Belgium) software. To evaluate the correlation between the clinical pathological characteristics and *TERRA* levels in tissues or in plasma samples, the patients were divided into 2 groups with high and low *TERRA* levels by using R (RQ_HCC_/RQ_PT_) = 0.7 for tissue expression or Ct = 22.81 (the median value) for plasma levels. Fisher’s exact test was applied. Data were considered statistically significant when *p*-value ≤ 0.05.

## Figures and Tables

**Figure 1 ijms-23-06183-f001:**
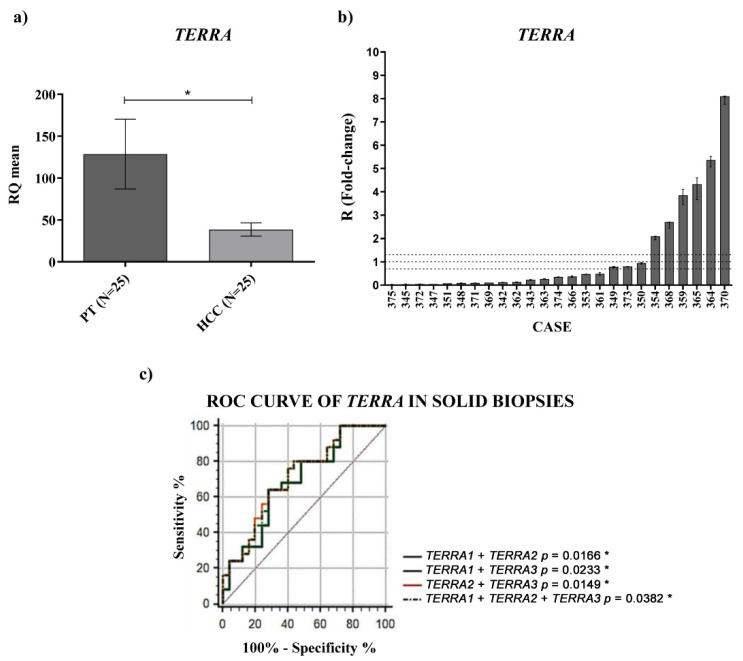
Tissue expression of *TERRA*. (**a**) *TERRA* relative quantification mean comparison between PT and HCC. Wilcoxon test was used. (**b**) *TERRA* fold-change. Histograms represent R-values and bars upper and lower limits. Histograms are ordered by increasing R-values. The corresponding cases are listed on x axis. Upper and lower dot lines define the range of gene expression variation. (**c**) ROC curve analysis of *TERRA* in tissues. Sensitivity: percent of correctly classified HCC patients. Specificity: percent of correctly classified non-HCC. AUC: area under the ROC curve; CI: confidence interval. * *p* < 0.05.

**Figure 2 ijms-23-06183-f002:**
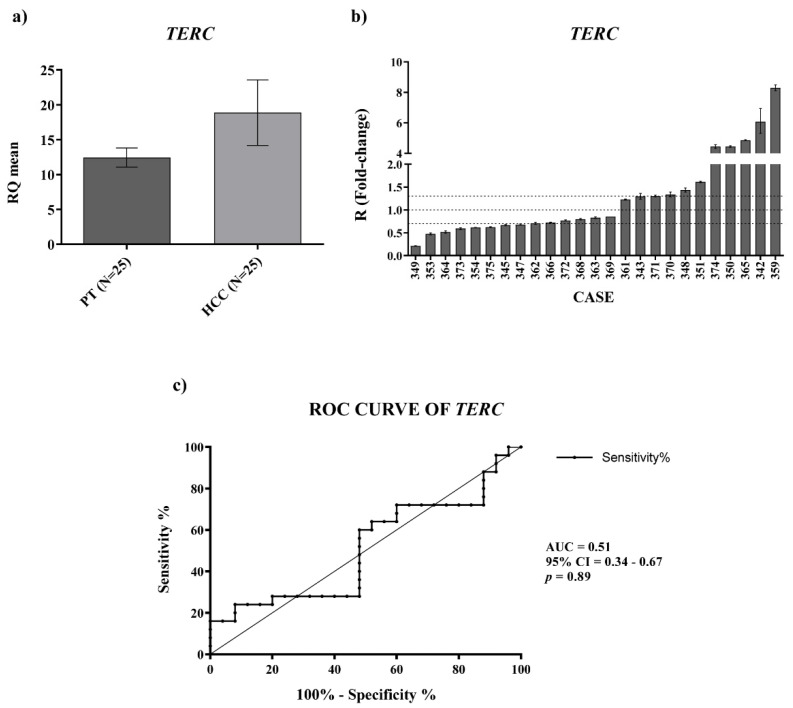
Tissue expression of *TERC*. (**a**) *TERC* relative quantification mean comparison between PT and HCC. Wilcoxon test was used. (**b**) *TERC* fold-change. Histograms represent R-values and bars upper and lower limits. Histograms are ordered by increasing R-values. The corresponding cases are listed on x axis. Upper and lower dot lines define the range of gene expression variation. (**c**) ROC curve analysis of *TERC* to discriminate HCC from normal tissue. AUC: area under the ROC curve’ CI: confidence interval.

**Figure 3 ijms-23-06183-f003:**
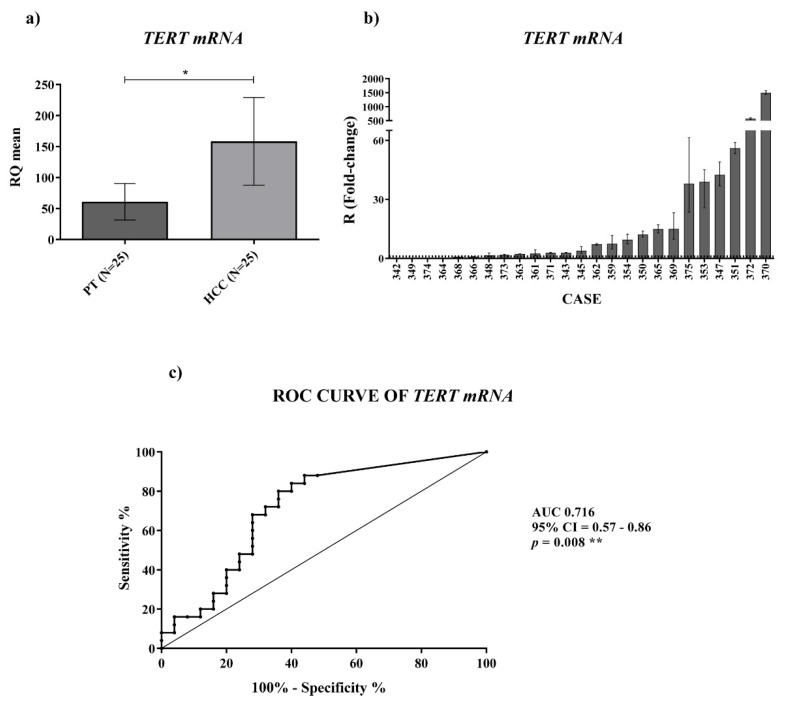
Tissue expression of *TERT* mRNA. (**a**) *TERT* mRNA relative quantification mean comparison between PT and HCC. Wilcoxon test was used. (**b**) *TERT* mRNA fold-change. Histograms represent R-values and bars upper and lower limits. Histograms are ordered by increasing R-values. The corresponding cases are listed on x axis. Upper and lower dot lines define the range of gene expression variation. (**c**) ROC curve analysis to discriminate HCC from normal tissue. AUC: area under the ROC curve; CI: confidence interval. * *p* < 0.05; ** *p* < 0.01.

**Figure 4 ijms-23-06183-f004:**
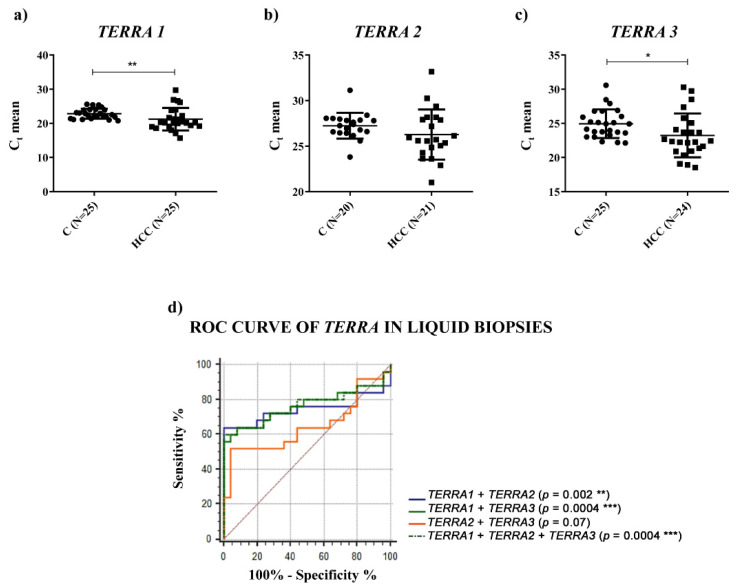
Plasma levels of *TERRA*. (**a**–**c**) *TERRA* levels in terms of C_t_ in plasma of healthy (C) and HCC individuals. *TERRA* expression from different telomers was analyzed using specific primers (*TERRA* 1 = hTel 1_2_10_13q; *TERRA* 2 = hTel 15q; *TERRA* 3 = hTel XpYp). Parametric unpaired two-tailed *t*-test. (**d**) ROC curve analysis of *TERRA* to discriminate HCC from healthy individuals. * *p* < 0.05, ** *p* < 0.01, *** *p* < 0.001.

**Figure 5 ijms-23-06183-f005:**
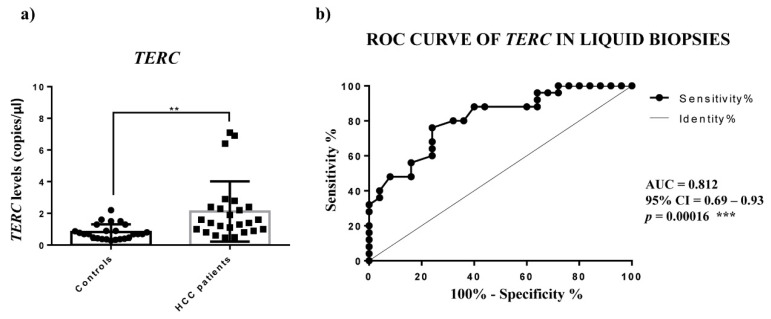
Plasma levels of *TERC*. (**a**) *TERC* levels in terms of copies/µL in plasma of control subjects (C) and HCC patients. Unpaired *t*-test was used. (**b**) ROC curve analysis of *TERC* to discriminate HCC from healthy individuals. ** *p* < 0.01; *** *p* < 0.001.

**Figure 6 ijms-23-06183-f006:**
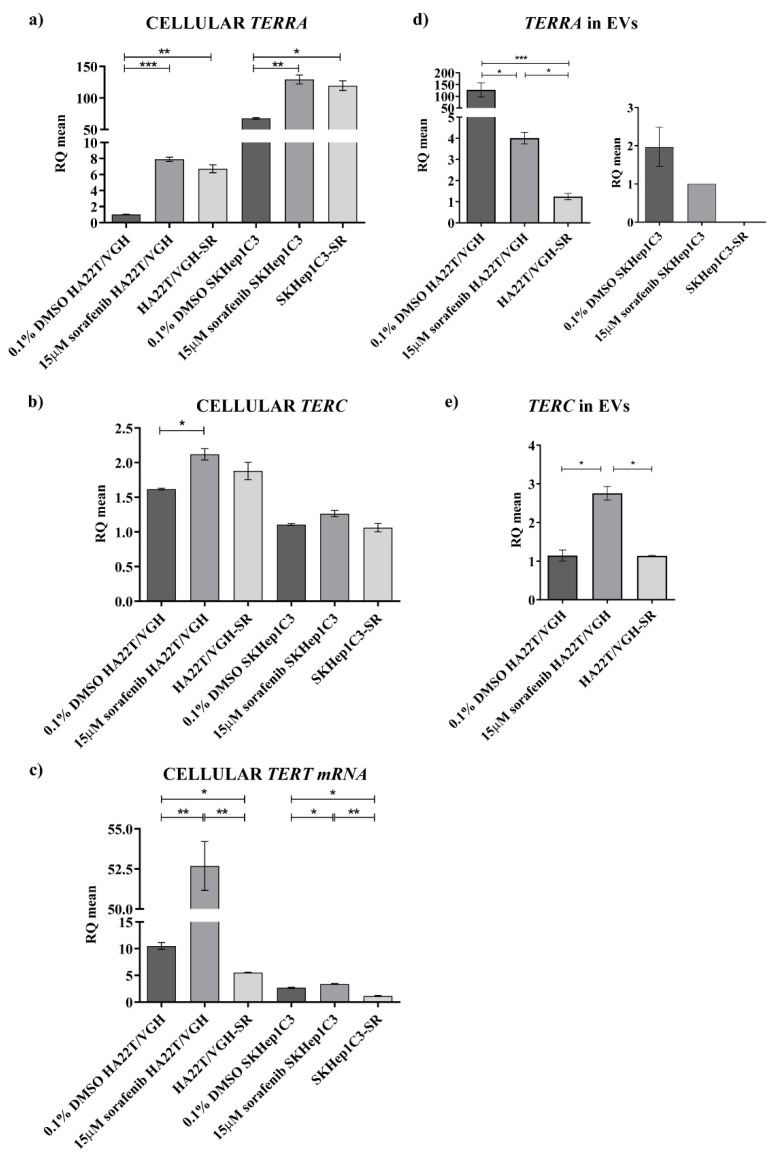
Intra- and extracellular expression of *TERRA* and telomerase transcripts in HCC cell lines. The expression levels of *TERRA* (**a**), *TERC* (**b**) and *TERT* mRNA (**c**) were measured in HA22T/VGH and SKHep1C3 untreated, resistant to sorafenib (SR) and treated with 15 µM sorafenib. The levels of *TERRA* (**d**) and *TERC* (**e**) were measured in EVs from HA22T/VGH and SKHep1C3 cells untreated, resistant to sorafenib (SR) and treated with 15 µM sorafenib. As described in Materials and Methods, gene expression level was normalized to the number of particles released per each biological sample, respectively. The histograms represent the mean values of 2 experiments, bars are ± SEM. Unpaired two tailed *t*-test; * *p* < 0.05, ** *p* < 0.01, *** *p* < 0.001.

**Table 1 ijms-23-06183-t001:** Diagnostic performance of *TERRA* in HCC tissues.

TARGET	Sensitivity	Specificity	AUC	SE	95% CI	*p*-Value
TERRA1	80%	64%	0.706	0.075	0.560–0.826	0.0064 **
TERRA2	84%	52%	0.698	0.074	0.551–0.819	0.008 **
TERRA3	84%	52%	0.712	0.072	0.567–0.831	0.0036 **
TERRA1+TERRA2	52%	80%	0.686	0.075	0.540–0.810	0.0166 *
TERRA1+TERRA3	52%	80%	0.71	0.072	0.565–0.830	0.0233 *
TERRA2+TERRA3	44%	80%	0.712	0.072	0.567–0.831	0.0149 *
TERRA1+TERRA2+TERRA3	44%	80%	0.709	0.073	0.563–0.829	0.0382 *

* *p* < 0.05; ** *p* < 0.01.

**Table 2 ijms-23-06183-t002:** Diagnostic performance of circulating *TERRA* in plasma.

TARGET	Sensitivity	Specificity	AUC	SE	95% CI	*p*-Value
TERRA1	60%	100%	0.747	0.079	0.604–0.859	0.001 **
TERRA2	52%	92%	0.622	0.084	0.474–0.756	0.146
TERRA3	56%	88%	0.686	0.08	0.540–0.810	0.02 *
TERRA1+TERRA2	64%	100%	0.742	0.081	0.599–0.856	0.002 **
TERRA1+TERRA3	64%	92%	0.763	0.074	0.622–0,872	0.0004 ***
TERRA2+TERRA3	52%	96%	0.648	0.083	0.500–0.778	0.07
TERRA1+TERRA2+TERRA3	60%	100%	0.765	0.074	0.624–0.873	0.0004 ***

*^*^ p* < 0.05; ** *p* < 0.01; *** *p* < 0.001.

## Data Availability

The data supporting the findings of this study are contained within the contents of this article. The datasets generated during this study will be freely provided by the corresponding author upon request.

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
