# Peer review of "Expression of Cellular and Extracellular TERRA, TERC and TERT in Hepatocellular Carcinoma"

_ijms, 2022, doi:10.3390/ijms23116183_

Round 1
Reviewer 1 Report
Manganelli et al., report the dysregulation of TERT and non-coding RNA TERRA and TERC levels in hepatocellular carcinoma. While TERRA mRNA reduces in HCC tissues, circulating TERRA in plasma increases. Circulating TERC is also increased in HCC patients. The authors also show that TERRA can be secreted in EVs from HCC cell lines. These observations are interesting and may be applied for non-invasive diagnostic molecular indicators of HCC. Some questions should be addressed before publication.
Major comments
- Is any correlation between the low TERRA levels in the solid biopsies and the high levels of circulating TERRA from individual HCC patients? Please conduct statistical analysis for this.
- It seems that there is a negative correlation between cellular TERRA and global TERRA in EVs (Fig 6) after sorafenib treatment. It is unclear how small TERRA in EV was produced. The sentence should be rephrased in Line 208 “a high fold increase of cellular TERRA that is secreted into the EVs, but to a lesser extent”. The cellular TERRA (long TERRA) detected here was not the same TERRA (small TERRA) in EVs.
- Whether high levels of TERRA and TERC in plasma are specific for HCC or also can be found in other cancer types? This could be discussed.
Minor comments
- Please label the P values of ROC curves in Figures 1c and 4d.
- Line 200, the TERRA sized detected in the EVs was smaller than 200 nt or 200~500 nt? Please show the Northern data.
- Please describe the full name of the Receiver Operating characteristic (ROC) curve in the abstract or Line98 when you mentioned it first in the article.
Author Response
Major comments
- Is any correlation between the low TERRA levels in the solid biopsies and the high levels of circulating TERRA from individual HCC patients? Please conduct statistical analysis for this. We thank the reviewer for this request. We evaluated the correlation by Spearman’s analysis of TERRA levels between the HCC tissues and the plasma samples from the same patients. The r-value was 0.3096 and the P-value was not significant (P-value=0.1410). Thus, in our cohort there was no correlation between TERRA levels in HCC solid biopsies and liquid biopsies. This analysis has been included in the revised version of the manuscript as new supplementary Figure (Figure S3).
- It seems that there is a negative correlation between cellular TERRA and global TERRA in EVs (Fig 6) after sorafenib treatment. We appreciate this consideration. Actually, we found an opposite trend between the expression levels of cellular TERRA and the amounts contained in the EVs after sorafenib treatment in the 2 cell lines considered. It is unclear how small TERRA in EV was produced. See the results obtained by northern blotting analysis. The sentence should be rephrased in Line 208 “a high fold increase of cellular TERRA that is secreted into the EVs, but to a lesser extent”. We changed and simplified the sentence: “In particular, these findings reveal that the HCC cells expressing low TERRA levels (HA22T/VGH) show a very high TERRA amount into the EVs. Anyway, in both cell lines, the sorafenib treatment leads to a cellular TERRA increase and at the same time causes a TERRA decrease in the EVs. The cellular TERRA (long TERRA) detected here was not the same TERRA (small TERRA) in EVs. We used the same primer pairs to detect cellular TERRA and TERRA contained in EVs. To avoid misunderstanding we have deleted the words “forms of TERRA”.
- Whether high levels of TERRA and TERC in plasma are specific for HCC or also can be found in other cancer types? This could be discussed. We thank the reviewer for this comment and request. We have answered this question in the discussion reporting these informations. For TERRA transcripts in the plasma, a RNA-seq study [18] revealed the presence of TERRA molecules in the plasma of 2 normal subjects and of 19 patients with various cancer type (prostate, stomach, breast, colon, kidney, lung, duct, liver, melanoma, ovarian); but, to our knowledge, there are no reports on circulating TERRA in a small or large cohort of patients with the same type of cancer; even no reports on the PCR determinations of TERRA amounts in the plasma of cancer patients. Concerning TERC in cancer, it is known from in situ and fluorescence hybridization data (ISH, FISH ) that is overexpressed in lung, oral, prostate and cervix carcinoma [10]; further only Novakovic et al , to our knowledge, reported nested-PCR data that revealed the presence of TERC in the plasma of healthy controls and of patients with breast cancer, melanoma and thyroid cancer, but did not allow to consider the TERC amount as tumor marker [31]. Hence, our findings would indicate the determination of circulating TERRA and TERC in HCC patients as specific non-invasive molecular indicators of this malignant condition.
Minor comments
- Please label the P values of ROC curves in Figures 1c and 4d.
As requested, we have included the P-values and the asterisks in the legend of the figures.
2. Line 200, the TERRA sized detected in the EVs was smaller than 200 nt or 200~500 nt? Please show the Northern data. The TERRA size detected in the EVs was smaller than 500 nt as revealed by Northern analysis (Figure S4).
3. Please describe the full name of the Receiver Operating characteristic (ROC) curve in the abstract or Line98 when you mentioned it first in the article.
In the revised version of the manuscript, we have described the full name of the ROC both in the abstract and in line 98.
Reviewer 2 Report
This manuscript examines the expression levels of selected non-coding and coding transcripts in hepatocellular carcinoma (HCC) known to be important for telomere homeostasis in cancer. In vivo and in vitro material was studied; in vivo: tumor and corresponding peritumoral tissue, EDTA plasma from HCC patients and healthy controls; in vitro: two HCC tumor cell lines, two sensitive and two established as resistant to a single chemotherapeutic agent (sorafinib). The number (n=25) of clinical cases appears rather small, as does the small number (n=2) of tumor cell lines used for this descriptive cohort study. The following list of concerns indicates why this manuscript cannot be recommended for publication:
- This is an observational and descriptive study without any functional aspect, although this would be possible with the cell lines included, but not by treatment with an agent acting through unresolved mechanisms. Because of this fact and the small number of cases and models studied, the findings on possible mechanisms are speculative. In addition, all reported in vivo observations are based on studies of bulk cells by qPCR and do not allow conclusions about single cells, e.g. tumor cells, in general.
- The main findings as described in the title, the abstract, the first and last paragraph of the discussion are not consistent, and it is not clear what the message of this report should be.
- Observations that primarily TERRA is downregulated and TERT is upregulated have been published for many cancers, including HCC, as indicated in the references described by the authors. Reference #2 contains such data and should be added to this list (line 222).
- Observations of elevated TERRA levels in serum of HCC patients are based on assays without appropriate controls and are reported as Ct values only. Further experiments are needed to substantiate these results and exclude artifacts.
- TERRA qPCR assays used do not cover all possible transcripts from all chromosomal ends, see e.g. reference #31. How can the authors conclude in Fig. 1 that TERRA transcripts are generally screened globally? Furthermore, recent publications have shown that certain genomic loci are responsible as the main source of human TERRA transcripts; has this fact been taken into account? No information is provided on the normalization procedure used to calculate RQ values, so it is not clear whether RQ values can be compared between TERRA1-3 PCR assays.
- Authors describe a comparison with clinicopathological characteristics without further explaining which data and how (line 232).
- Table S1 describes 26 HCC patients and does not match the number described (line 302).
- Northern analyses should be included and shown (line 200).
Author Response
- This is an observational and descriptive study without any functional aspect, although this would be possible with the cell lines included, but not by treatment with an agent actingthrough unresolved mechanisms. Because of this fact and the small number of cases and models studied, the findings on possible mechanisms are speculative. In addition, all reported in vivo observations are based on studies of bulk cells by qPCR and do not allow conclusions about single cells, e.g. tumor cells, in general. The main findings as described in the title, the abstract, the first and last paragraph of the discussion are not consistent, and it is not clear what the message of this report should be. We thank the reviewer for these comments. In the revised version of the manuscript, we deleted the term “dysregulation” in the title and in the abstract. Now the title is “Expression of Cellular and Extracellular Non Coding Telomeric-Repeat-Containing RNAs in Hepatocellular Carcinoma”. The first and last paragraphs of the discussion have been modified in order to make it more consistent and to better define the message of the manuscript.
- Observations that primarily TERRA is downregulated and TERT is upregulated have been published for many cancers, including HCC, as indicated in the references described by the authors. Reference #2 contains such data and should be added to this list (line 222). The reference #2 has been included in the list of the references cited in line 222.
- Observations of elevated TERRA levels in serum of HCC patients are based on assays without appropriate controls and are reported as Ct values only. In our study, TERRA levels were determined in plasma not in serum. The results of these experiments were reported as Ct average since no endogenous transcript was established as normalizer for plasma. Several published studies on circulating RNA levels reported the qPCR results as mean Ct values, as we did. Further experiments are needed to substantiate these results and exclude artifacts. We evaluated the circulating levels of TERRA transcribed from different chromosome loci (TERRA1:1_2_10-13q; TERRA2:15q; TERRA3: XpYp). We plotted the results as mean Ct for TERRA1, TERRA2 and TERRA3 by testing the samples in triplicate. As shown in Fig. 4a, 4b, 4c, we detected higher levels in plasma from HCC patients respect to the plasma from healthy controls. In support of these findings and to exclude artifacts, we calculated the average Ct values (named TERRA) in the 2 groups of individuals and we obtained significant higher levels in plasma from HCC patients respect to healthy control subjects. These results have been included in the new supplementary Figure S2 of the revised version of the manuscript (Figure S2).
5. TERRA qPCR assays used do not cover all possible transcripts from all chromosomal ends, see e.g. reference #31. How can the authors conclude in Fig. 1 that TERRA transcripts are generally screened globally? We thank the reviewer for this comment. Like in other published papers [27, 28, 35), we used a qPCR assay that did not detect all TERRA transcripts but it detected the most abundant ones. In detail, we used 3 primer pairs to quantify TERRA expression levels transcribed from 7 telomeres (TERRA1: 1_2_10-13q; TERRA2: 15q; TERRA3: XpYp). Furthermore, Diman et al. (2016) [29] demonstrated that the primer pair named TERRA1 (1_2_10_13q) detected the transcripts expressed from the telomeres: 1_2_4_10_13_22q. Thus according to Diman et al., our qPCR assay should recognize TERRA transcripts expressed from 9 telomeres. Anyway, in the revised version of the manuscript, we omitted the terms “global/globally” and TERRA is used to indicate the expression levels of the all transcripts evaluated; TERRA1, TERRA2, TERRA3 are used for the expression levels derived from the specific loci. Furthermore, recent publications have shown that certain genomic loci are responsible as the main source of human TERRA transcripts; has this fact been taken into account? It has been reported that human TERRA is mainly transcribed from the short arm of the chromosome X (Xp) subtelomere [30]. We included in our qPCR the primer pairs able to detect these transcripts. No information is provided on the normalization procedure used to calculate RQ values, so it is not clear whether RQ values can be compared between TERRA1-3 PCR assays. As reported in Materials and Methods section, TERRA cDNA pool and β-actin cDNA was synthesized from 1 µg of total RNA form tissues and cells by using specific RT-primers. qPCR amplifications were performed using primer pairs listed in lines 393-398. β-actin was used as reference gene to calculate the relative quantification (RQ) of TERRA1, TERRA2 and TERRA3 by using the 2-ΔΔCt method [36]. Thus, according to the described procedure we can conclude that the RQ values can be compared between TERRA1-3.
6. Authors describe a comparison with clinicopathological characteristics without further explaining which data and how (line 232). We thank the referee for this clarification. In the revised version of the manuscript we have included new supplementary Tables (Table S1 and Table S2) reporting the association analysis between the clinicopathological characteristics of HCC patients and TERRA expression levels in solid and liquid biospies. The analysis has been performed by Fisher’s exact test and non-significant correlations were obtained. We included the analysis in the M&M and the Results sections.
7. Table S1 describes 26 HCC patients and does not match the number described (line 302). As reported in the footnotes of the Table S3 for the case 352 and the case 371 only peripheral blood and solid biopsy were available respectively.
8. Northern analyses should be included and shown (line 200). The results obtained by Northern blot analysis have been included in the new supplementary Figure S4 and the results are described and discussed in the text.
Round 2
Reviewer 2 Report
The manuscript has not been sufficiently improved with regard to several points raised:
Point 1: Why TERT/TERC are not mentioned in title?
Point 5: The included reference #30 reports not Xp but 20q subtelomere as a main TERRA locus in human cells. The information described by the authors should be corrected accordingly. Regarding normalisation of RQ values, only the reference gene BACT but not the reference sample is described. This information is needed to conclude that the RQ values can be compared.
Point 7: Table S1 lists 26 different case numbers that do not match with the described total 25 HCC patients resected (line 416 of v2.pdf)
Author Response
The manuscript has not been sufficiently improved with regard to several points raised:
Point 1: Why TERT/TERC are not mentioned in title?
We thank the referee for this request and very gladly we accept the proposal to include TERC and TERT in the title. The new title is: “Expression of Cellular and Extracellular TERRA, TERC and TERT in Hepatocellular Carcinoma”
Point 5: The included reference #30 reports not Xp but 20q subtelomere as a main TERRA locus in human cells. The information described by the authors should be corrected accordingly. We thank the referee for the clarification and, as requested, we have corrected this information in the Discussion section.
Regarding normalisation of RQ values, only the reference gene BACT but not the reference sample is described. This information is needed to conclude that the RQ values can be compared.
We thank the reviewer for this consideration. Regarding normalization of RQ values, we performed qPCR analysis as described. Each cDNA from PT and HCC tissue samples was amplified by qPCR in triplicate. In detail, in each 96-well plate, 4 patients at a time (hence, 4 PT and 4 HCC tissues) were assessed for the expression levels of β-actin, TERRA1, TERRA2, TERRA3 (totally 96 wells). The quality check on Ct standard deviation and melting curve analysis was performed. Therefore data analysis was conducted using the 2^(-ΔΔCt) method (ref. #34). When all the 25 patients were analyzed, the plates were virtually merged by the 7500 Applied Biosystems software in order to determine the same calibrator per target (TERRA1, TERRA1, TERRA3) that was the sample with the lowest TERRA expression (in our cohort the sample HCC347). For all these reasons, RQ values of the different samples can be compared.
Point 7: Table S1 lists 26 different case numbers that do not match with the described total 25 HCC patients resected (line 416 of v2.pdf)
We apologize for not being sufficiently clear on this point in the previous response to the referee n. 2.
The total number of patients recruited was 26. However, for patient n. 352 we collected only the peripheral blood and for patient n. 371 we collected only the solid biopsy (as indicated in the footnotes of the Table S3). Figures 1, 2, 3 refer to the expression of TERRA, TERC and TERT respectively and the patient n. 352 was NOT included among the HCC solid biopsies analyzed (n=25). Similarly, Figures 4-5 refer to circulating TERRA and TERC levels respectively and the patient n. 371 was NOT included among the liquid biopsies of patients with HCC (n=25).
Fig. S1 reports the R values for each target (TERRA1, TERRA2, TERRA3) determined in solid biopsies from HCC patients (n=25), and the patient n. 352 was NOT included.
Fig. S2 shows the Ct values of TERRA levels in plasma from healthy individuals (n=25) and HCC patients (n=25) and the patient n. 371 was NOT included.
Fig. S3 shows the correlation between circulating levels of TERRA in plasma of HCC patients and the expression levels in their matched solid biopsies. The total number of cases analyzed was n=24 because the patients n. 352 and 371 were excluded from this analysis.
Finally, Tab S1 shows the association analysis between the clinicopathological characteristics of HCC patients and the TERRA expression levels in solid biopsies, and the patient n. 352 was not included. Similarly, the Table S2 reports the association analysis between the clinicopathological characteristics of HCC patients and the circulating levels of TERRA in HCC patients, and the patient n. 371 was excluded.